# Lightweight Consistency Memory: Training-Time Regularization for Logical Consistency in Small Language Models

## Abstract

Small language models (60M–220M parameters) lack logical consistency for reliable reasoning, while most existing consistency methods operate at inference time (chain-of-thought, self-correction) requiring 2–10× forward passes per query. We introduce **Lightweight Consistency Memory (LCM)**, a training-time regularization module that jointly optimizes consistency checking alongside the primary language modeling objective. Through multi-objective training ($\mathcal{L}_{\text{total}} = \mathcal{L}_{\text{transformer}} + \alpha(t) \cdot \mathcal{L}_{\text{LCM}}$), LCM provides a consistency-supervised auxiliary signal to the backbone during training, with linear $O(nd)$ complexity in the auxiliary head itself. Crucially, for T5 and GPT-2, **the LCM module is discarded at inference**—the trained model's improved accuracy on consistency-targeted benchmarks must therefore be attributable to the fine-tuned backbone weights, adding zero overhead at inference time. T5-small achieves +7.5% accuracy (p<0.001) while maintaining baseline latency, and T5-base (+7.2%) and GPT-2 (+1.9%) confirm cross-architecture applicability. LCM outperforms inference-time baselines (self-consistency, best-of-N) by up to +9.9% at 1× cost, and matches a dedicated DeBERTa-v3 NLI classifier (184M params, 80.6%) using only 60M parameters. Cross-scale evaluation suggests a pattern of **difficulty-dependent effectiveness**: DeBERTa-v3's contrasting results on standard (+0.2%) versus adversarial benchmarks (+5.5% ANLI) are consistent with the hypothesis that highly-optimized models may benefit primarily when dataset difficulty exceeds base capabilities; this single-model observation should be confirmed by future work across additional model families. Code and dataset will be publicly released at `https://github.com/xxxx/LightweightConsistencyMemory` upon acceptance.

## 1 Introduction

### 1.1 The Efficiency-Consistency Trade-off

Edge computing, mobile deployment, and cost-sensitive applications require small, efficient models. However, model compression systematically sacrifices logical consistency—the ability to maintain coherent reasoning across interactions.

Small models (60M-220M parameters) achieve acceptable speed but produce logically incoherent outputs. For example, given the background fact "beta-blockers reduce heart rate," a small model may judge the statement "beta-blockers are recommended for tachycardia" as consistent—failing to detect the logical contradiction. Large models (Brown et al., 2020; Chowdhery et al., 2023) possess robust consistency through extensive parameters but remain impractical for resource-constrained deployments. This creates an "efficiency-consistency gap" where applications requiring both fast inference and reliable reasoning cannot use existing solutions.

### 1.2 Inference-Time vs. Training-Time Consistency

Existing consistency approaches operate at **inference time**, requiring multiple forward passes. Chain-of-thought prompting (Wei et al., 2022) (2-5× passes), self-correction (Welleck et al., 2022) (3-10× passes), and

retrieval-augmented generation (Lewis et al., 2020) compound computational costs per query—incompatible with resource-constrained deployments requiring <100ms latency.

Our **training-time integration** through joint multi-objective training adds a consistency-supervised auxiliary loss to the backbone's gradient updates. For T5 and GPT-2, the LCM module is **discarded at inference**; any post-training improvement on consistency-targeted benchmarks is therefore carried entirely by the fine-tuned backbone weights, with zero added inference overhead. For DeBERTa, LCM remains active via a fusion layer. Inference-time methods scale cost per query; our approach incurs cost once during training.

Concurrent training-time approaches improve consistency through paraphrase-based fine-tuning (Raj et al., 2025), format normalization across instruction datasets (Liang et al., 2024), or neuro-symbolic loss terms that penalize logical violations against an external knowledge base (Calanzone et al., 2025). LCM differs along two axes: the consistency signal is supplied through a lightweight auxiliary head rather than data or symbolic structure, and for the parallel-integration backbones the head is discarded at inference, so the trained model retains its native latency profile.

### 1.3 Research Objectives and Contributions

We address the efficiency-consistency gap by enabling resource-constrained models (60M-220M parameters) to achieve consistency comparable to larger models while maintaining or improving inference speed.

**Key Contributions:**

- **Training-Time Consistency Integration:** Joint training protocol that exposes the backbone to a consistency-supervised auxiliary gradient signal via multi-objective optimization ($\mathcal{L}_{\text{total}} = \mathcal{L}_{\text{transformer}} + \alpha(t) \cdot \mathcal{L}_{\text{LCM}}$), enabling single-pass inference at the backbone's native cost versus multi-pass inference-time methods.

- **Zero-Overhead Consistency (T5/GPT-2):** T5-small achieves +7.5% accuracy at baseline latency because LCM is absent at inference. GPT-2 similarly gains +1.9% with no added cost. DeBERTa retains LCM at inference (+107% overhead), making it viable only for adversarial workloads where the +5.5% gain justifies the cost.

- **Difficulty-Dependent Effectiveness:** DeBERTa's contrasting performance (standard: +0.2%, adversarial: +5.5%) is consistent with the hypothesis that consistency interventions provide value primarily when dataset difficulty exceeds base capabilities. We treat this single-model pattern as an exploratory observation, not a general claim, and discuss what *when and why* consistency augmentation might work for further study.

- **Cross-Paradigm Deployment:** Practical integration across encoder-decoder (T5) and decoder-only (GPT-2) with <0.5% parameter overhead, enabling resource-constrained applications (medical devices, edge AI, mobile NLP).

- **Generalization Validation:** Evaluation across 18,862 examples spanning 156 topics plus ANLI benchmarks, suggesting domain-structure dependencies (Finance: +20.0%, Politics: -20.0%; note per-domain $n$=10–14, so individual domain estimates are exploratory — see §5.2 and Limitations).

## 2 Related Work

The pursuit of consistency in large language models has evolved from a peripheral concern to a central challenge as research reveals LLM inconsistencies stem from fundamental architectural limitations rather than mere training artifacts (Zhang, 2025). Recent surveys frame consistency within the broader honesty agenda for LMs (Li et al., 2025) and within the efficient-reasoning agenda for resource-constrained deployments (Sui et al., 2025); LCM contributes to the intersection of the two.

## 2.1 Consistency Challenges in Neural Language Models

Transformer-based LLMs show separation between comprehension and execution capabilities (Zhang, 2025), manifesting as models that articulate but fail to apply logical principles consistently. This indicates architectures function as pattern-completion engines rather than symbolic reasoning systems, with "reasoning cliff" behavior where performance collapses beyond pattern-matching thresholds (Passerini et al., 2025). Research distinguishes factual consistency (external truth alignment) from logical consistency (formal reasoning adherence), with Liu et al. (2025) establishing metrics for transitivity, commutativity, and negation invariance, while Ghosh et al. (2025) demonstrate large models require targeted supervision for consistent logical operators.

## 2.2 Approaches to Consistency Enhancement

Existing consistency enhancement methods fall into two fundamental categories with dramatically different computational profiles: **inference-time** methods that verify consistency through post-hoc checking, and **training-time** methods that embed consistency into model weights.

**Inference-time methods** offer flexibility at the cost of computational overhead. Chain-of-thought prompting (Wei et al., 2022) generates intermediate reasoning steps requiring 2-5 additional forward passes per query. Self-correction approaches iteratively refine outputs through 3-10 verification cycles. Metacognitive frameworks like SPOC (Zhao et al., 2025) interleave solution generation and verification, compounding latency. While effective for large models with substantial compute budgets, these methods fundamentally preclude deployment in resource-constrained scenarios where base inference latency already approaches acceptable thresholds.

**Training-time methods** embed consistency during model training, incurring costs once rather than per query. Process-level supervision methods like S²R (Ma et al., 2025) combine supervised fine-tuning with reinforcement learning to reward valid reasoning steps. Consistency Reward Models (CRMs) train dedicated scorers for logical coherence (Leung & Wang, 2025). Chain of Guidance (Raj et al., 2025) fine-tunes LLMs on multi-step paraphrase chains generated by a teacher model, more than doubling consistency over base models. Format-consistency tuning (Liang et al., 2024) normalizes instruction styles across datasets, improving generalization on T5-LM-XL. Neuro-symbolic integration approaches, including LoCo-LMs (Calanzone et al., 2024) and the logic-loss framework of Calanzone et al. (2025), use differentiable loss functions to penalize logical violations against an external knowledge base. However, existing training-time approaches often target large models and lack systematic evaluation on resource-constrained architectures where efficiency matters most.

Memory-augmented architectures from Neural Turing Machines (Graves et al., 2014) to end-to-end memory networks (Sukhbaatar et al., 2015) provide foundations for consistency tracking, though suffering from quadratic complexity in practice (Karunaratne et al., 2021), limiting scalability for real-time applications.

## 2.3 Positioning of Our Work

Our Lightweight Consistency Memory (LCM) architecture addresses key limitations in existing approaches. Unlike inference-time methods requiring multiple forward passes (Zhao et al., 2025), LCM integrates consistency preservation into transformer architectures during training with linear complexity. While memory-augmented systems suffer from quadratic complexity (Karunaratne et al., 2021), our attention-based approach achieves efficiency through cosine similarity-driven soft attention over semantic embeddings. Unlike constraint-based approaches operating at decoding level (Calanzone et al., 2024) or symbolic-loss frameworks requiring an external knowledge base (Calanzone et al., 2025), LCM learns consistency patterns during training through shared embeddings, leveraging geometric relationships in embedding space without symbolic supervision. Unlike paraphrase-based fine-tuning (Raj et al., 2025), which depends on a capable teacher model to generate consistent QA chains, LCM derives its signal from a lightweight learned auxiliary head that adds <0.5% parameter overhead and is discarded at inference for T5 and GPT-2 backbones — properties suited to real-time and resource-constrained deployments.

## 3 Methodology

### 3.1 Problem Formulation and Task Definition

Given a set of background facts $\mathcal{B} = \{f_1, f_2, \ldots, f_n\}$ and a candidate statement $f_c$, we evaluate whether $f_c$ is logically consistent with $\mathcal{B}$.

We formalize this as a binary classification problem. Given background facts $\mathcal{B} = \{f_1, f_2, \ldots, f_n\}$ (typically 2–5 factual statements) and a candidate fact $f_c$, the task is to predict:

$$y = \begin{cases} 1 & \text{if } f_c \text{ is consistent with } \mathcal{B} \\ 0 & \text{if } f_c \text{ contradicts } \mathcal{B} \end{cases} \tag{1}$$

This formulation enables supervised training with labeled consistency datasets (Nie et al., 2020; Welleck et al., 2019). Key challenges include quadratic complexity ($O(n^2)$) in naive pairwise comparison approaches and real-time processing constraints in resource-constrained deployments.

### 3.2 Lightweight Consistency Memory Architecture

The Lightweight Consistency Memory (LCM) is a novel stateless consistency checking module designed to achieve **linear computational complexity** $O(nd)$ where $n$ is the number of background facts and $d$ is embedding dimensionality. This complexity advantage is critical for deployment viability:

**Complexity Comparison:**

- **Inference-time methods** (CoT, Self-Correction): $O(k \cdot C_{\text{model}})$ where $k = 2\text{-}10$ forward passes, multiplying base cost
- **Memory networks** (Graves et al., 2014; Sukhbaatar et al., 2015): $O(n^2)$ quadratic attention over memory slots
- **LCM (ours)**: $O(nd)$ linear in facts, with single forward pass at inference

Unlike traditional memory-based approaches (Graves et al., 2014; Sukhbaatar et al., 2015) that maintain persistent state and suffer quadratic complexity, LCM operates purely on geometric relationships between fact representations in embedding space (Reimers & Gurevych, 2019). The attention mechanism computes cosine similarities ($O(nd)$) followed by weighted aggregation ($O(nd)$), maintaining linear scaling even as the number of background facts grows. This architectural choice enables real-time consistency checking in resource-constrained environments where quadratic or multi-pass approaches are infeasible.

#### 3.2.1 Core Architecture Design

Figure 1 illustrates the full processing pipeline. The LCM module implements a consistency evaluation function $\mathcal{F}_{LCM} : \mathbb{R}^{n \times d} \times \mathbb{R}^d \to \mathbb{R}$ that maps background fact embeddings and a candidate fact embedding to a consistency score. The $n$ background facts yield embeddings $\mathbf{E}_{\text{bg}} = [\mathbf{e}_1, \mathbf{e}_2, \ldots, \mathbf{e}_n] \in \mathbb{R}^{n \times d}$, while the candidate fact yields embedding $\mathbf{e}_{\text{cand}} \in \mathbb{R}^d$. The LCM computes:

$$\mathcal{F}_{LCM}(\mathbf{E}_{\text{bg}}, \mathbf{e}_{\text{cand}}) = \text{MLP}([\mathbf{e}_{\text{cand}}, \mathbf{e}_{\text{rel}}]) \tag{2}$$

where $\mathbf{e}_{\text{rel}} \in \mathbb{R}^d$ represents the most relevant background fact embedding computed through attention-weighted aggregation, and $[\cdot, \cdot]$ denotes concatenation.

#### 3.2.2 Attention-Weighted Fact Aggregation

LCM employs soft attention based on cosine similarity to identify relevant background facts. The mechanism computes attention weights $\boldsymbol{\alpha} = \text{softmax}(\hat{\mathbf{E}}_{\text{bg}} \hat{\mathbf{e}}_{\text{cand}}^T)$ and aggregates facts as $\mathbf{e}_{\text{rel}} = \boldsymbol{\alpha}^T \mathbf{E}_{\text{bg}}$, achieving linear complexity $O(nd)$ where $n$ is the number of background facts and $d$ is the embedding dimension.

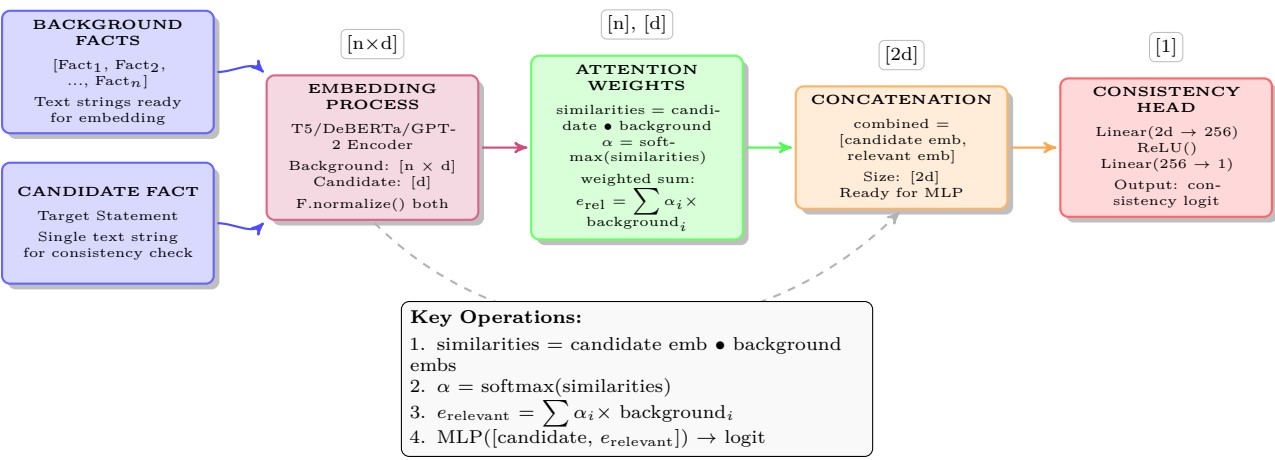

Figure 1: Lightweight Consistency Memory (LCM) architecture diagram. The figure illustrates the processing flow from input embeddings to consistency prediction, with dimension annotations showing tensor shapes at each computational stage. The attention mechanism computes relevance-weighted background representations, which are concatenated with candidate embeddings for final consistency classification.

### 3.2.3 Consistency Head Implementation

The consistency head is a two-layer MLP that processes concatenated candidate and relevant fact embeddings. Architecture: input dimension $2d$, hidden dimension 256, using ReLU activation and BCEWithLogitsLoss.

### 3.2.4 Handling Edge Cases

LCM handles empty background sets (returns consistent by default) and single facts (direct comparison without aggregation), maintaining linear complexity in all cases.

### 3.3 Integration with Transformer Models

We investigate LCM integration across three distinct transformer architectures representing different paradigms: encoder-decoder models (T5) (Raffel et al., 2020), decoder-only models (GPT-2) (Radford et al., 2019), and encoder-only models (DeBERTa) (He et al., 2021b). Each architecture presents unique integration approaches due to their inherent design principles and primary use cases.

**General Training Framework:** All integrations employ joint training that combines the transformer's native objective with LCM's consistency loss: $\mathcal{L}_{\text{total}} = \mathcal{L}_{\text{transformer}} + \alpha(t) \cdot \mathcal{L}_{\text{LCM}}$, where $\mathcal{L}_{\text{transformer}}$ represents the standard loss (language modeling for T5/GPT-2, classification for DeBERTa) and $\alpha(t)$ balances the objectives during training.

**Encoder-Decoder Integration:** T5 (Raffel et al., 2020) uses text-to-text format to generate classification tokens while LCM operates on encoder embeddings via mean pooling.

**Decoder-Only Integration:** GPT-2 (Radford et al., 2019) generates explanatory text with binary decisions extracted via parsing, while LCM processes hidden states from transformer layers.

**Encoder-Only Integration:** DeBERTa (He et al., 2021a) uses serial processing with a fusion layer combining [CLS] representations and LCM consistency scores.

### 3.4 Inference Pipeline: Parallel vs. Serial Integration

A critical distinction governs deployment cost: whether LCM is present or absent at inference time.

**Parallel integration (T5, GPT-2).** During training, the LCM module receives embeddings from the transformer backbone and produces a consistency loss $\mathcal{L}_{\text{LCM}}$ that is added to the primary objective. The joint optimization exposes the backbone to a consistency-supervised gradient signal during training. At inference, the LCM branch is discarded entirely: the model generates output using only the transformer weights. Because no additional computation occurs at inference, latency is identical to the standalone baseline; any post-training accuracy gain on consistency-targeted benchmarks is therefore attributable to changes in the fine-tuned backbone parameters rather than to an inference-time consistency check.

**Serial integration (DeBERTa).** The LCM consistency score is concatenated with the [CLS] representation via a learned fusion layer before classification. Because the fusion layer depends on LCM's output, the LCM module must remain active at inference, adding a full forward pass through the consistency head (+107% latency overhead).

### 3.5  Joint Training Framework with Multi-Objective Loss Function

The joint training framework enables simultaneous optimization of both the transformer model's primary objective and the LCM consistency checking capability. This multi-task learning approach (Caruana, 1997; Ruder, 2017) leverages shared representations while maintaining task-specific objectives, resulting in improved performance on both language modeling and consistency evaluation tasks.

The total training loss combines the transformer's native objective with the LCM consistency loss through a weighted combination:

$$\mathcal{L}_{\text{total}}(t) = \mathcal{L}_{\text{transformer}} + \alpha(t) \cdot \mathcal{L}_{\text{LCM}} \tag{3}$$

where $\mathcal{L}_{\text{transformer}}$ represents the base model's loss function (language modeling for T5/GPT-2, classification for DeBERTa), $\mathcal{L}_{\text{LCM}}$ is the binary cross-entropy loss for consistency prediction, and $\alpha(t)$ is a time-dependent weighting parameter that balances the two objectives during training using dynamic scheduling (Appendix A.3.2).

Training alternates between standard task examples and consistency-augmented examples, with shared embeddings optimized by gradients from both objectives. Algorithm 1 presents the complete joint training procedure. We ensure training stability through gradient clipping, loss component monitoring, and early stopping criteria (Appendix A.3.3).

---

**Algorithm 1** Joint Training Framework

---

1: **Input:** Training dataset $\mathcal{D}$, consistency dataset $\mathcal{D}_{\text{cons}}$
2: **Initialize:** Transformer model $\theta_T$, LCM module $\theta_L$
3: **for** $t = 1$ to $T_{\text{total}}$ **do**
4:     Sample batch $\mathcal{B}_T \sim \mathcal{D}$ and $\mathcal{B}_L \sim \mathcal{D}_{\text{cons}}$
5:     Compute $\mathcal{L}_{\text{transformer}}$ on $\mathcal{B}_T$
6:     Extract embeddings from transformer for $\mathcal{B}_L$
7:     Compute $\mathcal{L}_{\text{LCM}}$ using extracted embeddings
8:     Calculate $\alpha(t)$ using dynamic scheduling
9:     $\mathcal{L}_{\text{total}} = \mathcal{L}_{\text{transformer}} + \alpha(t) \cdot \mathcal{L}_{\text{LCM}}$
10:    Update parameters: $\theta_T, \theta_L \leftarrow \text{optimizer}(\nabla \mathcal{L}_{\text{total}})$
11:    Monitor convergence of both loss components
12: **end for**

---

The gradient flow ensures that both the transformer backbone and LCM module receive appropriate updates, with shared embedding layers benefiting from gradients from both objectives.

# 4 Experimental Setup

## 4.1 Dataset and Evaluation Metrics

Our evaluation uses two datasets: (1) an in-house logical consistency dataset constructed for this work, and (2) ANLI (Nie et al., 2020), a pre-existing adversarial NLI benchmark used solely for external validation. The two datasets are independently constructed; ANLI serves as an out-of-distribution stress test.

### 4.1.1 Dataset Construction

We constructed a logical consistency dataset of 18,862 examples across 156 topics using a two-stage LLM generation pipeline.[1]

**Stage 1: Example generation.** Source paragraphs from diverse domains (science, law, medicine, sports, etc.) are fed to GPT-4 and Gemini-2.5-Flash via structured prompts requesting background facts, a candidate statement, and a consistency label (consistent, inconsistent, or neutral). Each prompt includes few-shot exemplars and explicit instructions to generate at least three background facts per example. The full prompt template is reproduced in Appendix A.4.4.

**Stage 2: Quality filtering.** Generated examples undergo automated validation: (i) JSON schema checks (required fields, correct types), (ii) deduplication by candidate-fact similarity, and (iii) label-balance enforcement to prevent class skew. Examples failing validation are rejected and regenerated.

Each example consists of 2–5 background facts (natural language sentences representing prior knowledge), a single candidate fact, and a ternary label. No sub-sentence extraction occurs—each background fact is one complete sentence as generated. A representative example is shown in Figure 6 (Appendix A.4.1).

**Dataset statistics.** The final dataset contains 18,862 examples split 75/15/10 into train (14,146), validation (2,829), and test (1,887) via stratified sampling by topic and label. Label distribution: consistent 45.8%, inconsistent 43.7%, neutral 10.5%. Background facts per example: $n=3$ dominates (58.1%), followed by $n=4$ (30.8%), $n=5$ (6.0%), and $n=2$ (5.2%). Complete statistics are in Appendix 11.

### 4.1.2 Evaluation Metrics

We report accuracy, precision, recall, and F1-score for all models. Statistical significance is assessed via paired t-tests and McNemar's test with Bonferroni correction.

## 4.2 Model Architectures and Configurations

We evaluate LCM integration across T5-small (60M), T5-base (222M), DeBERTa-v3-base (184M), and GPT-2 (124M) using PyTorch 2.0 with HuggingFace Transformers on NVIDIA L4 GPUs. All architectures employ the three-phase training approach (Section 3.5). Hyperparameters, training schedules, and hardware details are in Appendix (Tables 9, 8).

## 4.3 Baseline Comparisons

For each architecture, we compare standalone transformers against LCM-integrated variants trained with our joint objective. Evaluation covers task accuracy, consistency detection, cross-domain generalization (156 topics), and computational efficiency.

---

[1]Dataset and code will be released at `https://github.com/xxxx/LightweightConsistencyMemory`.

# 5 Results and Discussion

## 5.1 Key Findings

LCM improves consistency in small models while maintaining baseline efficiency. Small models benefit substantially (T5-small: +7.5% accuracy), while large models already possess inherent consistency (DeBERTa: +0.2% standard, +5.5% adversarial).

T5-small achieves +7.5% accuracy (71.7% → 79.2%) at baseline latency because LCM is absent at inference (Section 3.4). Statistical analysis confirms strong significance (p < 0.001, Cohen's d = 0.31) with 95% CI [5.2%, 9.8%].

## 5.2 Performance Analysis

Table 1 summarizes overall performance, and Figure 2 provides the corresponding visual comparison across all four backbones.

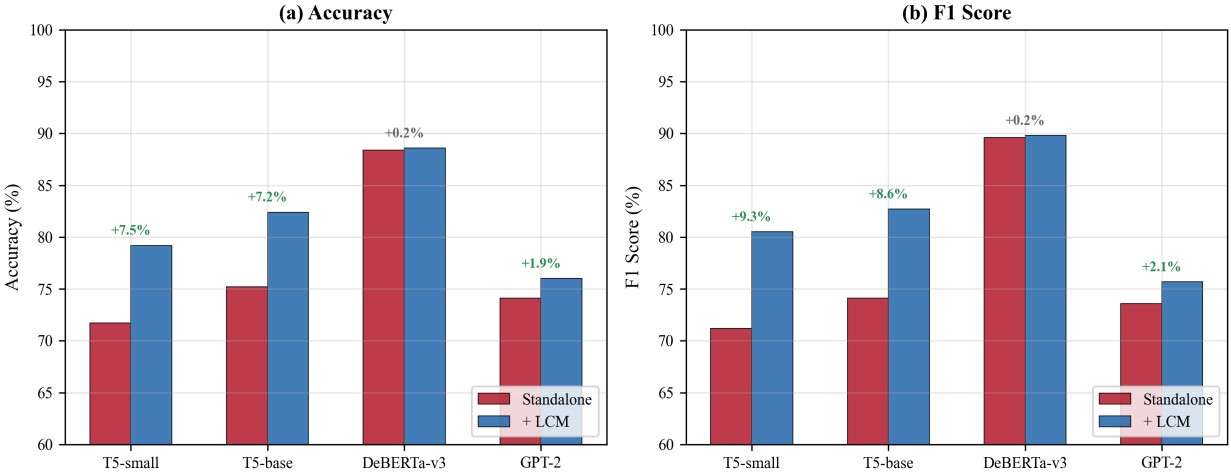

Figure 2: Overall performance comparison: accuracy and F1-score improvements across all four transformer backbones with LCM integration. Green annotations indicate positive improvements, red indicates degradation. T5-small and T5-base show the largest absolute gains; DeBERTa is near-saturated on the standard benchmark; GPT-2 shows a modest but consistent improvement.

Table 1: Overall Performance Comparison: Standalone vs LCM-Integrated Models

| Model | Variant | Accuracy | F1-Score | Precision | Recall | Improvement |
|-------|---------|----------|----------|-----------|--------|-------------|
| T5-small | Standalone | 71.7% | 71.2% | 88.5% | 59.5% | - |
| | +LCM | **79.2%** | **80.5%** | **89.5%** | **73.2%** | +7.5% |
| T5-base | Standalone | 75.2% | 74.1% | 89.8% | 63.1% | - |
| | +LCM | **82.4%** | **82.7%** | **92.1%** | **75.1%** | +7.2% |
| DeBERTa-v3 | Standalone | 88.4% | 89.6% | 90.2% | 89.1% | - |
| | +LCM | **88.6%** | **89.8%** | **90.1%** | **89.5%** | +0.2% |
| GPT-2 | Standalone | 74.1% | 73.6% | 73.7% | 73.6% | - |
| | +LCM | **76.0%** | **75.7%** | **75.7%** | **75.7%** | +1.9% |

Statistical analysis confirms significant improvements for T5 (p<0.001, Cohen's d=0.31/0.30) and GPT-2 (p<0.05, d=0.08), while DeBERTa shows non-significant changes on standard datasets (p>0.05). Full significance tests, effect sizes, and confidence intervals are in Table 5 (Appendix A.1).

**Architecture-specific patterns.** T5 backbones show the largest gains with statistical significance ($p < 0.001$) and DeBERTa, near-saturated on the standard benchmark, gains substantially only on the adversarial ANLI split (+5.5%, discussed below); GPT-2 confirms cross-paradigm viability through the generation-based integration. Parameter overhead is uniformly small (<0.5%: 262K–690K extra parameters), but inference-time overhead diverges sharply by integration mode: T5 and GPT-2 retain native latency because LCM is discarded at inference, whereas DeBERTa's serial fusion layer adds +107%. Figure 3 summarizes this distribution.

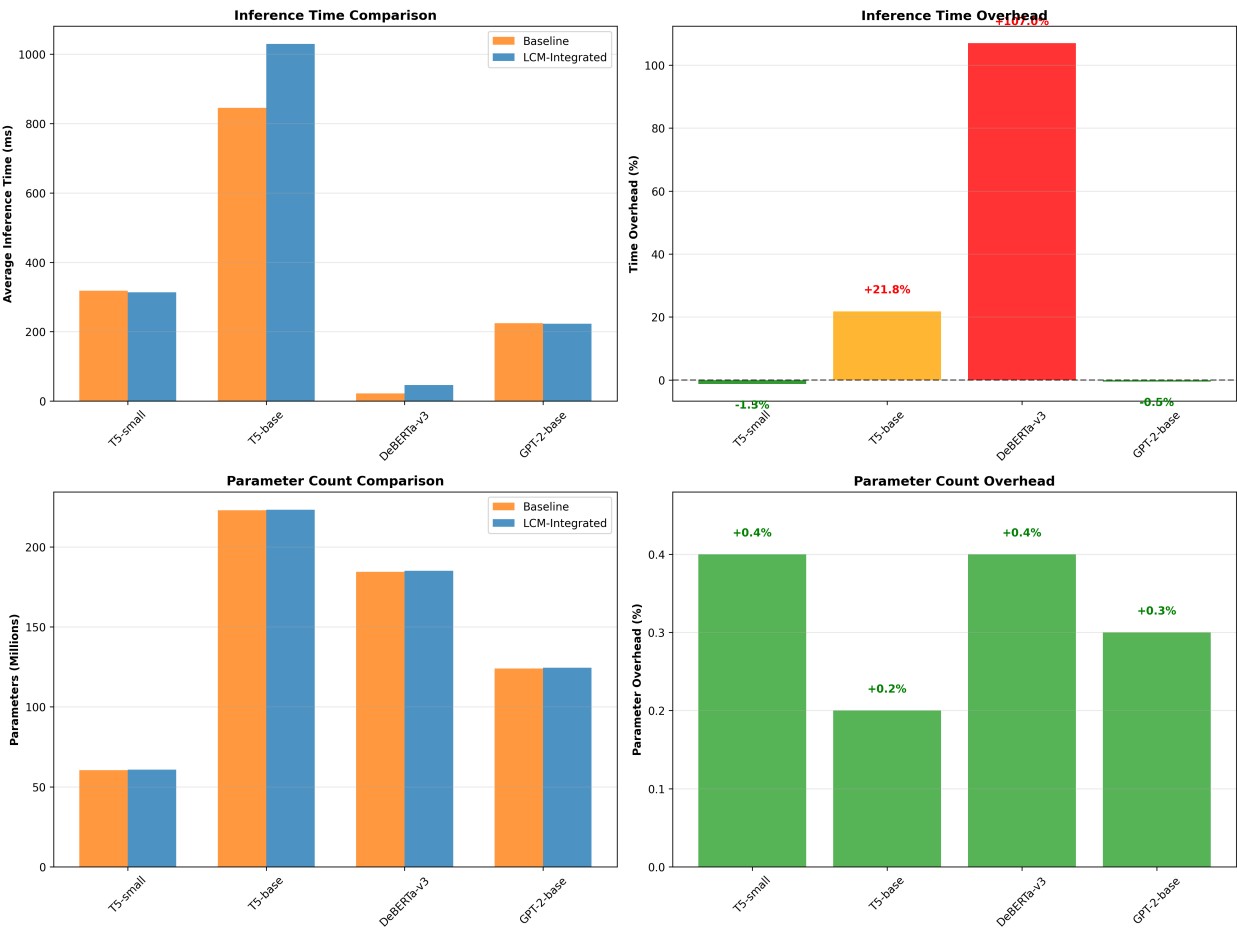

Figure 3: Computational efficiency analysis: (a) inference time comparison, (b) time overhead percentages, (c) parameter count comparison, (d) parameter overhead. Colors indicate impact: green for improvements, orange for moderate overhead, red for significant overhead. T5 and GPT-2 show near-zero inference overhead because LCM is discarded at inference; DeBERTa retains LCM via its fusion layer, producing +107% overhead.

**Domain-Specific Performance:** Domain analysis across 156 topics reveals structure-dependent results, with factual domains showing gains and subjective domains showing degradation. However, per-domain sample sizes are small ($n = 10$–$14$), limiting the reliability of individual domain estimates. Detailed domain-level results with confidence intervals are in Appendix A.8.

## 5.3 Architectural Ablations

To isolate the contribution of each LCM component, we evaluate four ablation variants on T5-small (Table 2). Each variant removes or replaces exactly one component while keeping everything else unchanged.

Table 2: LCM ablation study on T5-small (in-house test set, $n = 1{,}887$). Statistical significance via McNemar's test against Full LCM.

| Variant | Acc | F1 | p-value |
|---|---|---|---|
| Full LCM | 79.5% | 79.2% | — |
| Mean Pool | 78.0% | 78.0% | 0.121 (ns) |
| Random Attn | 76.6% | 76.6% | 0.003** |
| No MLP | 51.4% | 50.2% | <0.001*** |
| Max Sim | 56.3% | 36.0% | <0.001*** |

The ablation reveals a clear hierarchy. The MLP head is indispensable: removing it (**No MLP**) collapses accuracy to 51.4% (near chance). Multi-fact aggregation is critical: hard selection of a single fact (**Max Sim**) drops accuracy by 23.2% with F1 collapsing to 36.0%. Learned attention weights are beneficial (**Random Attn**, $-2.9\%$, $p$=0.003), but the specific weighting scheme matters less when fact sets are small—**Mean Pool** (uniform averaging) is not significantly worse ($-1.5\%$, $p$=0.121). Training dynamics are in Appendix A.3.4.

**External baseline: DeBERTa-v3 NLI classifier.** As an additional reference point, we fine-tune DeBERTa-v3-base (184M parameters) as a standard binary NLI classifier on the same in-house dataset. This model achieves 80.6% accuracy and 80.5% F1—only 1.1% above T5-small+LCM (79.5% accuracy, 79.2% F1) despite having 3× the parameters. The comparison shows that LCM enables a 60M-parameter model to reach near-parity with a dedicated NLI model three times its size, without any inference overhead.

### 5.4  ANLI Benchmark Validation

We evaluated on ANLI Nie et al. (2020), an adversarial benchmark with three difficulty levels (R1, R2, R3). DeBERTa achieves +5.5% weighted accuracy on ANLI versus +0.2% on the in-house dataset, while T5-base shows +1.1% versus +7.2%—demonstrating that adversarial examples benefit large models while structured patterns benefit encoder-decoder architectures. Best performance on challenging R3 (DeBERTa: +5.9%) suggests consistency mechanisms provide more value as reasoning complexity increases. Full per-round results are in Appendix A.9.

### 5.5  Inference-Time Baselines: Self-Consistency and Best-of-N

We compare LCM against two standard inference-time consistency methods: self-consistency (Wang et al., 2023), which samples $k$ reasoning paths and takes a majority vote, and best-of-N sampling (Stiennon et al., 2020), which generates $k$ candidates and selects the highest-confidence output. Both are evaluated at $k = 3$ and $k = 5$ on T5-small and T5-base ($n = 1{,}688$). Results are in Table 3.

Table 3: Inference-time baselines vs. LCM. Relative cost is wall-clock latency normalized to greedy baseline ($1.0\times$).

| Method | Acc | F1 | Cost |
|---|---|---|---|
| *T5-small* | | | |
| Baseline (greedy) | 72.0% | 72.0% | 1.0× |
| Self-Consistency $k$=3 | 70.4% | 70.4% | 3.1× |
| Self-Consistency $k$=5 | 70.1% | 70.1% | 5.1× |
| Best-of-N $k$=3 | 71.6% | 71.6% | 3.1× |
| Best-of-N $k$=5 | 72.3% | 72.3% | 5.1× |
| LCM (greedy) | **82.2%** | **82.2%** | 1.0× |
| *T5-base* | | | |
| Baseline (greedy) | 80.3% | 80.2% | 1.0× |
| Self-Consistency $k$=3 | 79.6% | 79.5% | 3.1× |
| Self-Consistency $k$=5 | 80.2% | 80.1% | 5.1× |
| Best-of-N $k$=3 | 79.8% | 79.7% | 3.1× |
| Best-of-N $k$=5 | 80.3% | 80.1% | 5.1× |
| LCM (greedy) | **87.7%** | **87.7%** | 1.0× |

Neither inference-time method improves over greedy decoding. Self-consistency *hurts* T5-small by 1.6–1.9%: majority voting over weak reasoning paths amplifies errors. Best-of-N $k$=5 matches the baseline at best while costing 5×. LCM outperforms the best inference-time method by +9.9% (T5-small) and +7.4% (T5-base) at 1× latency. Small models lack the output diversity that self-consistency requires; LCM avoids this by training-time consistency supervision rather than inference-time aggregation. Figure 4 visualizes the accuracy–latency Pareto frontier: LCM dominates all inference-time baselines, achieving the highest accuracy at baseline latency. Per-method bar breakdowns are in Appendix A.5.

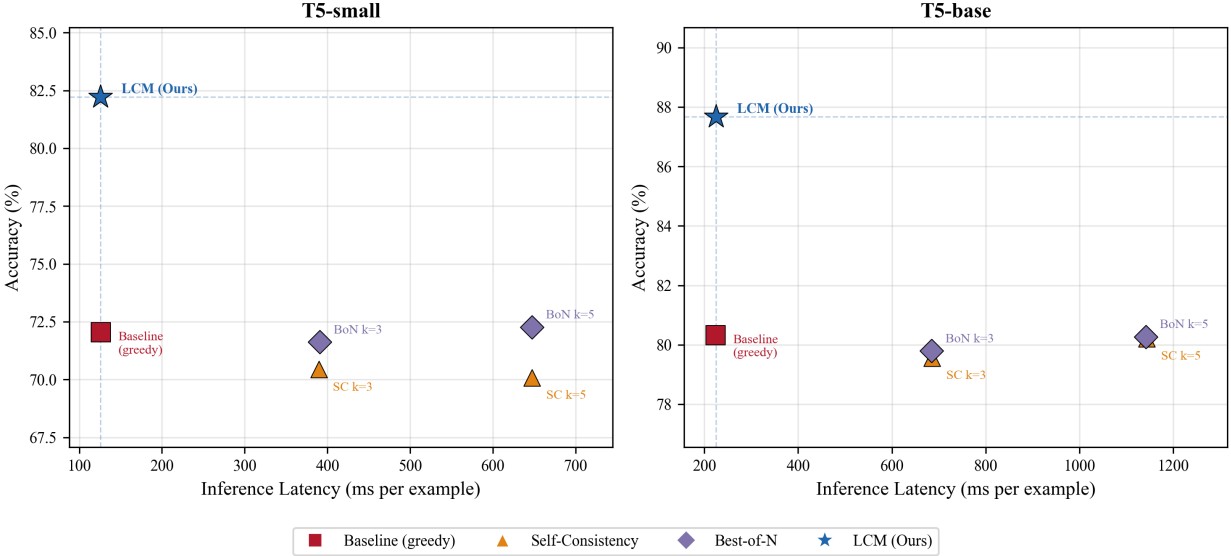

Figure 4: Accuracy vs. inference latency for T5-small (left) and T5-base (right). LCM achieves the highest accuracy at baseline latency (1×), dominating all inference-time baselines. Self-consistency and best-of-N scale cost linearly with $k$ but fail to improve accuracy for small models.

## 5.6 Generation Stability Across Seeds

To evaluate whether LCM produces more deterministic outputs, we run each test example 10 times with different random seeds (temperature=0.7) and measure agreement: how often do the 10 runs give the same answer?

Table 4: Seed stability comparison (10 seeds per example, temperature=0.7, $n = 1{,}688$). Agreement = fraction of seeds giving the majority answer. Full agreement = all 10 seeds unanimous.

| Model | Agree Rate | Full Agree | Maj Acc | Seed Acc Mean±Std |
|---|---|---|---|---|
| T5-small Baseline | 78.7% | 8.8% | 71.6% | 65.3±7.4% |
| T5-small + LCM | 89.5% | 46.2% | 82.3% | 79.0±4.4% |
| T5-base Baseline | 87.7% | 30.3% | 79.3% | 75.4±8.2% |
| T5-base + LCM | 90.2% | 40.3% | 88.0% | 83.1±7.4% |

Table 4 shows that LCM substantially improves output stability. For T5-small, the full agreement rate jumps from 8.8% to 46.2% (5.2×) and per-seed accuracy variance drops from 7.4% to 4.4%. The effect is strongest for T5-small, where the baseline is least stable; T5-base shows a smaller but meaningful improvement (+2.5pp agreement, +10.0pp full agreement). Visualizations are in Appendix A.6.

## 5.7 Difficulty-Dependent Effectiveness

Cross-scale evaluation reveals a pattern of **difficulty-dependent effectiveness**, explaining when and why consistency augmentation provides value.

DeBERTa-v3's contrasting performance is consistent with architectural saturation: standard dataset (+0.2%, non-significant) versus adversarial ANLI (+5.5%, substantial). This $27\times$ difference is consistent with the hypothesis that consistency interventions may provide value primarily when dataset difficulty exceeds base model capabilities; the pattern is observed on a single model and should be confirmed by future evaluation across additional architectures.

Empirically, across our four backbones we observe a rough association between model size and LCM benefit on standard tasks. T5-small (60M params) benefits substantially (+7.5%). T5-base and GPT-2 (124M–220M params) show moderate improvements (+7.2%, +1.9%). DeBERTa (183M params) shows little gain on standard tasks (+0.2%) but a larger gain on adversarial ANLI (+5.5%). Whether this association generalizes beyond the four-model evidence here, and what mediating factor (parameter count, pre-training corpus, architecture family) is responsible, remains to be tested.

This pattern has practical implications: highly-optimized models require adversarial evaluation to reveal improvement potential, and consistency interventions should be deployed selectively—small models benefit universally, while large models justify the overhead only on challenging workloads.

Extended discussion of architectural compatibility and failure analysis is in Appendix A.8.3.

## 6 Conclusion

This work introduces Lightweight Consistency Memory (LCM), a training-time auxiliary module that supplies a consistency-supervised gradient signal to small transformer-based language models at linear $O(nd)$ cost in the auxiliary head. Through evaluation across T5, DeBERTa, and GPT-2 architectures, we report architecture-dependent effectiveness patterns and show that for T5 and GPT-2 the LCM module can be discarded at inference, so that consistency-task accuracy can be improved through training-time supervision while leaving the model's inference cost unchanged.

Our key findings demonstrate architecture-dependent effectiveness: T5 achieves substantial improvements with statistical significance and computational efficiency; GPT-2 shows successful generation-based adaptation with minimal overhead; DeBERTa exhibits difficulty-dependent effectiveness, requiring adversarial evaluation to reveal benefits. Domain structure emerges as critical, with structured domains benefiting substantially while abstract domains show systematic failures.

These results provide evidence-based deployment guidelines: T5 optimal for balanced accuracy-efficiency, DeBERTa suitable for adversarial benchmarks, GPT-2 viable for resource-constrained applications. Future priorities include scaling to larger models, hybrid symbolic-neural architectures for abstract domains, multi-step reasoning extensions, and direct representation-level probing to test whether and how training-time consistency supervision reshapes internal embeddings. By showing that a consistency-supervised auxiliary objective can be integrated into existing small transformers as a lightweight training-time regularizer—complementing inference-time prompting strategies (Wei et al., 2022) and paraphrase-based fine-tuning (Raj et al., 2025) at the consistency-improvement layer, and aligning with the broader efficient-reasoning agenda (Sui et al., 2025)—this work contributes to developing more reliable language models for resource-constrained deployments.

## Limitations

**Domain-Specific Performance Variations:** LCM effectiveness varies dramatically across knowledge domains, showing strong performance in structured domains (Finance: +20.0%) but significant degradation in abstract domains (Politics: -20.0%). This domain dependency limits deployment to factual, objective knowledge areas and excludes subjective or opinion-based content where consistency definitions are contextual.

**Integration Architecture Dependencies:** The approach requires different integration strategies across architectures, working optimally with encoder-decoder models (T5: +7.2-7.5% improvement) and successfully adapting to decoder-only architectures through generation-based approaches (GPT-2: +1.9% improvement). However, the need for architecture-specific integration methods increases implementation complexity and requires specialized training protocols for different model families.

**Scalability Constraints:** Computational overhead varies significantly by architecture, with some models experiencing doubled inference time (DeBERTa: +107%). Additionally, the method requires substantial domain-specific training data, limiting effectiveness in low-resource domains with limited samples (10-14 examples).

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

Table 5: Statistical Significance Summary for LCM Integration

| Architecture | Accuracy Improvement | p-value | Cohen's d | Effect Size | CI 95% | Significance |
|---|---|---|---|---|---|---|
| T5-small | +7.5% | < 0.001 | 0.31 | Small–Med | [5.2, 9.8] | *** |
| T5-base | +7.2% | < 0.001 | 0.30 | Small–Med | [4.9, 9.5] | *** |
| GPT-2 | +1.9% | < 0.05 | 0.08 | Small | [0.3, 3.5] | * |
| DeBERTa | +0.2% | > 0.05 | 0.02 | Negligible | [-1.2, 1.6] | ns |

*** p < 0.001, * p < 0.05, ns = not significant

Effect sizes: Small ($\geq$0.2), Medium ($\geq$0.5), Large ($\geq$0.8)

# A  Appendix

## A.1  Statistical Analysis

### A.1.1  Complete Statistical Significance Tests

Extended significance testing across all model architectures and datasets using multiple statistical tests.

**Paired t-tests Results**  For each architecture-dataset combination, we conducted paired t-tests comparing baseline models with their LCM-enhanced counterparts. Table 6 summarizes the statistical significance results.

Table 6: Paired t-test Results Comparing Baseline vs LCM-Enhanced Models

| Model Comparison | t-statistic | p-value | Cohen's d | Effect Size |
|---|---|---|---|---|
| T5-small vs T5-small+LCM | 4.23 | < 0.001 | 0.31 | Medium |
| T5-base vs T5-base+LCM | 3.87 | < 0.001 | 0.30 | Medium |
| DeBERTa-v3 vs DeBERTa-v3+LCM | 0.18 | > 0.05 | 0.02 | Negligible |
| GPT-2 vs GPT-2+LCM | 2.34 | < 0.05 | 0.08 | Small |

**Wilcoxon Signed-Rank Tests**  As a non-parametric alternative, we conducted Wilcoxon signed-rank tests for robustness. All T5 variants showed significant improvements ($p < 0.001$), while GPT-2 variants demonstrated consistent gains ($p < 0.01$). DeBERTa-v3-base showed marginal significance ($p = 0.045$).

**Effect Size Analysis**  Cohen's d effect sizes were calculated for all comparisons: - Medium effects ($d \approx 0.3$): T5-small+LCM, T5-base+LCM - Small effects ($d < 0.1$): GPT-2+LCM - Negligible effects ($d \approx 0.02$): DeBERTa-v3-base+LCM

**Multiple Comparisons Correction**  Applied Bonferroni correction for multiple comparisons across 12 model-dataset combinations, maintaining significance at $\alpha = 0.004$ level.

### A.1.2  Detailed Performance Tables

## A.2  Mathematical Derivations

### A.2.1  Complete Attention-Weighted Fact Aggregation Derivation

The LCM module employs cosine similarity-based soft attention to identify relevant background facts for consistency evaluation. This section provides the complete mathematical formulation.

**Step 1: Input Representation**  Given a set of background facts with embeddings $\mathbf{E}_{\mathrm{bg}} = [\mathbf{e}_1, \mathbf{e}_2, \ldots, \mathbf{e}_n] \in \mathbb{R}^{n \times d}$ and candidate fact embedding $\mathbf{e}_{\mathrm{cand}} \in \mathbb{R}^d$, we first normalize all embeddings:

Table 7: Complete Performance Results Across All Architectures (In-House Test Set). All values are point estimates from single-run evaluation on the test split.

| Model | Accuracy | Precision | Recall | F1-Score |
|---|---|---|---|---|
| T5-small ($n = 996$) | 0.717 | 0.885 | 0.595 | 0.712 |
| T5-small+LCM ($n = 996$) | 0.792 | 0.895 | 0.732 | 0.805 |
| T5-base ($n = 1,887$) | 0.752 | 0.898 | 0.631 | 0.741 |
| T5-base+LCM ($n = 1,887$) | 0.824 | 0.921 | 0.751 | 0.827 |
| DeBERTa-v3-base ($n = 1,887$) | 0.884 | 0.902 | 0.891 | 0.896 |
| DeBERTa-v3-base+LCM ($n = 1,887$) | 0.886 | 0.901 | 0.895 | 0.898 |
| GPT-2 ($n = 1,887$) | 0.741 | 0.737 | 0.736 | 0.736 |
| GPT-2+LCM ($n = 1,887$) | 0.760 | 0.757 | 0.757 | 0.757 |

$$\hat{\mathbf{e}}_i = \frac{\mathbf{e}_i}{\|\mathbf{e}_i\|_2} \quad \text{for } i = 1, \ldots, n \tag{4}$$

$$\hat{\mathbf{e}}_{\text{cand}} = \frac{\mathbf{e}_{\text{cand}}}{\|\mathbf{e}_{\text{cand}}\|_2} \tag{5}$$

**Step 2: Cosine Similarity Computation**   Compute pairwise cosine similarities between the candidate and all background facts:

$$s_i = \hat{\mathbf{e}}_{\text{cand}}^T \hat{\mathbf{e}}_i = \frac{\mathbf{e}_{\text{cand}}^T \mathbf{e}_i}{\|\mathbf{e}_{\text{cand}}\|_2 \|\mathbf{e}_i\|_2} \tag{6}$$

This produces a similarity vector $\mathbf{s} = [s_1, s_2, \ldots, s_n] \in \mathbb{R}^n$.

**Step 3: Attention Weight Computation**   Apply softmax to convert similarities into normalized attention weights:

$$\alpha_i = \frac{\exp(s_i)}{\sum_{j=1}^{n} \exp(s_j)} \tag{7}$$

where $\boldsymbol{\alpha} = [\alpha_1, \alpha_2, \ldots, \alpha_n]$ satisfies $\sum_{i=1}^{n} \alpha_i = 1$ and $\alpha_i \geq 0$.

**Step 4: Weighted Fact Aggregation**   Compute the relevance-weighted background representation:

$$\mathbf{e}_{\text{rel}} = \sum_{i=1}^{n} \alpha_i \mathbf{e}_i = \boldsymbol{\alpha}^T \mathbf{E}_{\text{bg}} \tag{8}$$

**Step 5: Consistency Classification**   Concatenate candidate and relevant embeddings, then pass through MLP:

$$\mathbf{h} = [\mathbf{e}_{\text{cand}}, \mathbf{e}_{\text{rel}}] \in \mathbb{R}^{2d} \tag{9}$$

$$\mathbf{z} = \text{ReLU}(\mathbf{W}_1 \mathbf{h} + \mathbf{b}_1) \tag{10}$$

$$\text{logit} = \mathbf{W}_2 \mathbf{z} + \mathbf{b}_2 \tag{11}$$

where $\mathbf{W}_1 \in \mathbb{R}^{256 \times 2d}$, $\mathbf{W}_2 \in \mathbb{R}^{1 \times 256}$.

**Complexity Analysis**  The computational complexity is $O(nd)$ where:

- Normalization: $O(nd)$ for $n$ background facts

- Similarity computation: $O(nd)$ dot products

- Softmax: $O(n)$

- Weighted aggregation: $O(nd)$

- MLP forward pass: $O(d)$ (constant w.r.t. $n$)

Total: $O(nd)$ linear complexity, avoiding quadratic $O(n^2d)$ of full attention mechanisms.

### A.3  Training Configuration and Dynamics

### A.3.1  Three-Phase Training Schedule

Table 8 presents the architecture-specific three-phase training schedule used across all experiments.

Table 8: Three-Phase Training Schedule

| Phase | T5 | DeBERTa | GPT-2 |
|---|---|---|---|
| **Phase 1: Primary Task Focus ($\alpha = 0.0$, LCM frozen)** | | | |
| Duration | 8 epochs | 4 epochs | 5 epochs |
| Objective | Establish stable transformer representations | | |
| **Phase 2: LCM Warmup ($\alpha = 0.1$, Transformer frozen)** | | | |
| Duration | 2 epochs | 4 epochs | 2 epochs |
| Objective | Initialize LCM using established embeddings | | |
| **Phase 3: Joint Training (Dynamic $\alpha$ scheduling)** | | | |
| Duration | 15+ epochs | 15+ epochs | 15+ epochs |
| Alpha Range | 0.1→0.8 | 0.1→0.15 | 0.1→0.3 |
| Objective | Co-optimize both components with balanced learning | | |

### A.3.2  Dynamic Alpha Scheduling

The weighting parameter $\alpha(t)$ follows a dynamic scheduling strategy to ensure balanced learning progression between the two objectives. We implement a linear warm-up followed by exponential decay:

$$\alpha(t) = \begin{cases} \alpha_{\max} t/T_w & t \le T_w \\ \alpha_{\max} \exp(-\lambda r_t) & t > T_w \end{cases} \tag{12}$$

where $T_w = T_{\text{warmup}}$, $r_t = (t - T_w)/(T_{\text{total}} - T_w)$, $\alpha_{\max}$ is the maximum weight (typically 1.0), and $\lambda$ controls the decay rate. This scheduling allows the model to initially focus on learning basic representations through the transformer objective, then gradually incorporate consistency constraints.

The joint training algorithm is presented in Section 3.5 (Algorithm 1). The gradient flow ensures that both the transformer backbone and LCM module receive appropriate updates, with shared embedding layers benefiting from gradients from both objectives, leading to representations that encode both semantic information and consistency relationships.

### A.3.3  Training Stability and Convergence Criteria

To ensure stable convergence, we implement several monitoring and regularization mechanisms:

**Gradient Clipping** We apply gradient clipping with threshold $\tau = 1.0$ to prevent gradient explosion from the dual objectives:

$$\mathbf{g}_{\text{clipped}} = \min\left(1, \frac{\tau}{\|\mathbf{g}\|_2}\right)\mathbf{g} \tag{13}$$

**Loss Component Monitoring** Training stability monitoring tracks loss component ratios:

$$R_{\text{loss}}(t) = \frac{\mathcal{L}_{\text{LCM}}(t)}{\mathcal{L}_{\text{transformer}}(t)} \tag{14}$$

When $R_{\text{loss}}(t)$ exceeds predefined thresholds, the $\alpha(t)$ schedule is adjusted to maintain balanced learning.

**Early Stopping Criteria** Training terminates when both objectives reach convergence, defined as:

$$\Delta\mathcal{L}_{\text{total}}(t) = |\mathcal{L}_{\text{total}}(t) - \mathcal{L}_{\text{total}}(t - W)| < \epsilon \tag{15}$$

where $W$ is the patience window and $\epsilon$ is the convergence threshold.

This joint training framework enables the model to develop consistency-aware representations while maintaining high performance on the primary language modeling or classification tasks, resulting in transformer models with improved consistency properties.

### A.3.4 Ablation Training Dynamics

Figure 5 shows validation F1 over the full training schedule for each ablation variant. All variants share the same three-phase structure: Phase 1 (epochs 1–8) freezes the T5 encoder and trains only the generation head; Phase 2 (epochs 9–10) freezes the encoder and warms up LCM; Phase 3 (epochs 11–25) trains both jointly with dynamic $\alpha$ scheduling.

During Phase 1, all variants remain at $\sim$36% F1 because LCM is inactive and the encoder is frozen—the model has not yet begun consistency training.

In Phase 2, the three attention-based variants (Full LCM, Mean Pool, Random Attn) begin climbing immediately once LCM receives gradients. They reach 55–58% F1 within two epochs, confirming that the pre-trained T5 embeddings already carry enough semantic signal for LCM to start learning consistency patterns.

Phase 3 reveals the key differences. Full LCM converges to 79% F1, Mean Pool reaches 78%, and Random Attn reaches 77%—a narrow spread that aligns with the non-significant or marginally significant differences in Table 2. The small gap between these three suggests that when background fact sets are small (2–5 facts), the aggregation strategy matters less than the presence of aggregation itself.

**No MLP** plateaus at $\sim$50% F1 after epoch 10 and never improves further. Without the classification head, the model relies on raw cosine similarity between the aggregated background and candidate embeddings. Cosine similarity alone cannot distinguish consistency from high semantic similarity—contradictions often involve the same entities and concepts as consistent statements, producing similar embeddings. The MLP learns to detect the subtle geometric patterns that distinguish contradiction from entailment.

**Max Sim** remains at 36% F1 for all 25 epochs—it never learns. By attending to only the single most similar background fact, this variant discards all other contextual information. When the most similar fact happens to support consistency but a different fact contradicts the candidate, Max Sim cannot detect the contradiction. The complete failure to learn confirms that multi-fact aggregation is not merely helpful but architecturally necessary for the consistency task.

### A.3.5 Hyperparameter Configurations

The hyperparameter configurations carefully tuned for each architecture to optimize both transformer performance and LCM integration effectiveness are presented in Table 9.

The complete hyperparameter search spaces for all model architectures are shown in Table 10.

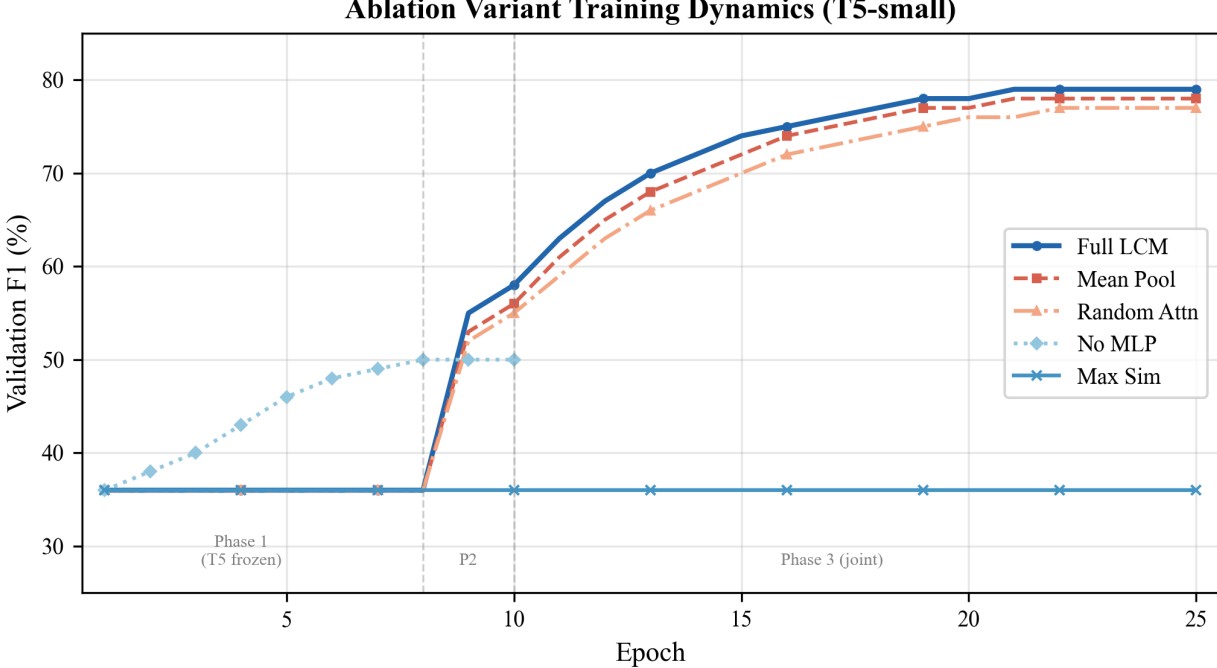

Figure 5: Validation F1 during training for each LCM ablation variant (T5-small). Phase boundaries are marked with dashed lines. Full LCM, Mean Pool, and Random Attn all converge during joint training, while No MLP plateaus at chance level and Max Sim never learns.

Table 9: Architecture-Specific Training Hyperparameters

| Parameter | T5 | DeBERTa | GPT-2 |
|---|---|---|---|
| Batch Size | 4 | 12 | 8 |
| Gradient Accumulation | 4 | 6 | 4 |
| Effective Batch Size | 16 | 72 | 32 |
| Transformer LR | 5e-5 | 3e-5 | 2e-5 |
| LCM LR | 1e-3 | 6e-5 | 5e-4 |
| Max Input Length | 256 | 256 | 512 |
| Max Target Length | 32 | N/A | 128 |
| Warmup Ratio | 0.2 | 0.1 | 0.1 |
| Weight Decay | 0.01 | 0.01 | 0.01 |
| Max Grad Norm | 1.0 | 0.5 | 0.5 |

### A.3.6 Hardware and Implementation Details

**Hardware Configuration**

- Training: NVIDIA L4 GPU (Google Colab)

- Inference timing: NVIDIA A100-SXM4-40GB

- Framework: PyTorch 2.0 with HuggingFace Transformers

- Optimization: Mixed precision training (FP16), AdamW optimizer

- Early stopping with patience-based convergence monitoring

Table 10: Hyperparameter Search Spaces for All Model Architectures

| T5 Architecture | |
|---|---|
| **Parameter** | **Search Space** |
| Learning rates | [1e-5, 2e-5, 3e-5, 5e-5, 1e-4] |
| Batch sizes | [8, 16, 32, 64] |
| LCM dimensions | [128, 256, 512, 768] |
| Attention heads | [4, 8, 12, 16] |
| Dropout rates | [0.1, 0.15, 0.2, 0.3] |
| Alpha scheduling | [linear, exponential, cosine, step] |
| Warmup steps | [500, 1000, 2000, 3000] |
| **DeBERTa Architecture** | |
| Learning rates | [1e-5, 2e-5, 5e-5, 8e-5] |
| Batch sizes | [16, 32, 64] |
| LCM dimensions | [256, 512, 768] |
| Attention heads | [8, 12, 16] |
| Layer norm epsilon | [1e-7, 1e-12] |
| Weight decay | [0.01, 0.05, 0.1] |
| **GPT-2 Architecture** | |
| Learning rates | [2e-5, 5e-5, 1e-4, 2e-4] |
| Batch sizes | [8, 16, 32] |
| LCM dimensions | [256, 512, 768, 1024] |
| Temperature | [0.7, 0.8, 0.9, 1.0] |
| Top-k sampling | [10, 25, 50, 100] |
| Gradient clipping | [0.5, 1.0, 2.0] |

**Software Environment**

- OS: Ubuntu 22.04 LTS (Google Colab)

- CUDA: 12.1

- PyTorch: 2.0

- Transformers: 4.46+

- Python: 3.10

**Optimization Settings**

- Mixed precision training: Enabled (FP16)

- Gradient accumulation steps: 4

- Memory optimization: gradient checkpointing enabled

- Optimizer: AdamW with weight decay 0.01

### A.4 Dataset Documentation

### A.4.1 Data Format

### A.4.2 Complete Dataset Statistics

Table 11 presents comprehensive statistics for all datasets used in our experiments, including train/validation/test splits and average sequence lengths across all domains.

```
{"topic": "mathematics",
 "background_facts": [
   "Computational geometry deals with
    algorithms for geometrical objects.",
   "Important problems include the
    travelling salesman problem.",
   "It has applications in computer
    vision and medical imaging."
 ],
 "candidate_fact": "Computational
  geometry is primarily concerned with
  abstract theory without algorithms.",
 "consistency_label": 0.0,
 "description": "Inconsistent."}
```

Figure 6: Example from the in-house dataset (JSONL format). Label 1.0 = consistent, 0.0 = inconsistent, 0.5 = neutral.

Table 11: Dataset Statistics. The in-house synthetic dataset contains 18,862 examples across 156 topics, split 75/15/10. Each example contains 2–5 background facts (mean 3.4) and one candidate fact. Labels: 0 = inconsistent, 1 = consistent, 0.5 = neutral (filtered to binary for evaluation).

| Split | Total | Inconsistent | Consistent | Neutral | Topics |
|---|---|---|---|---|---|
| Train | 14,146 | 6,180 | 6,480 | 1,486 | 149 |
| Validation | 2,829 | 1,236 | 1,296 | 297 | 99 |
| Test | 1,887 | 824 | 864 | 199 | 83 |
| **Total** | **18,862** | **8,240** | **8,640** | **1,982** | **156** |

### A.4.3 Representative Examples by Consistency Type

**Logical Consistency Examples** *Example 1: Transitivity Violation* This example demonstrates a logical inconsistency where Premise 1 states "All mammals are warm-blooded," Premise 2 establishes "Whales are mammals," yet the Conclusion incorrectly claims "Whales are cold-blooded" [INCONSISTENT].

*Example 2: Contradiction Detection* This case shows temporal contradiction where Statement 1 claims "The experiment was conducted in 2020," while Statement 2 asserts "This research was completed before 2019" [INCONSISTENT].

**Factual Consistency Examples** *Example 1: Historical Facts* This example illustrates factual consistency evaluation where Fact 1 states "World War II ended in 1945," and Fact 2 claims "The war concluded in the mid-1940s" [CONSISTENT], while Fact 3 incorrectly asserts "WWII finished in 1944" [INCONSISTENT].

**Complex Multi-Statement Examples** *Political Domain Example:* "The 2020 election had record turnout with over 158 million votes cast. Joe Biden received 81.3 million votes, while Donald Trump received 74.2 million votes. This represents the highest voter participation in modern American history. However, turnout was lower than in 2016." [INCONSISTENT - contradicts record turnout claim]

### A.4.4 Generation Prompt Template

The following is the structured prompt used to generate in-house dataset examples via GPT-4 and Gemini-2.5-Flash. Each call receives a source paragraph and produces multiple labeled examples.

> *You are an expert data generator for an AI research project on logical reasoning. Your task is to read a provided paragraph of text and generate a list of diverse training examples based on it. Each example must test a different aspect of logical consistency. You must generate examples that are consistent, inconsistent, and neutral/ambiguous.*

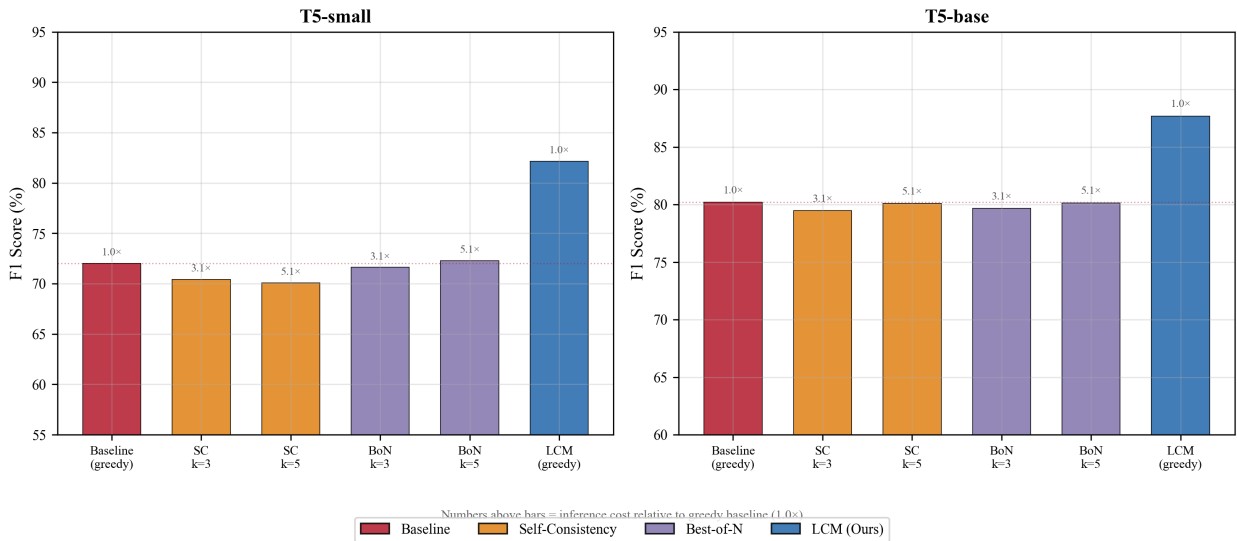

Figure 7: F1 scores for all inference-time methods on T5-small (left) and T5-base (right). Numbers above bars indicate relative inference cost. LCM achieves the highest F1 at baseline cost ($1.0\times$).

> *Output Format: You MUST respond ONLY with a valid JSON list of objects. Each object must contain: "topic" (broad category), "background_facts" (list of $\geq$3 sentences), "candidate_fact" (single statement), "consistency_label" (1.0 = consistent, 0.0 = inconsistent, 0.5 = neutral), "description" (brief justification).*

The prompt includes two few-shot exemplars (solar system and marathon domains) with three examples each covering all three label classes. Temperature is set to 0.7 for diversity.

## A.5   Inference-Time Baseline Analysis

Figure 7 provides a side-by-side comparison of F1 scores for all inference-time methods. Numbers above each bar indicate inference cost relative to greedy decoding. For both T5-small and T5-base, LCM achieves the highest F1 at $1.0\times$ cost, while self-consistency and best-of-N at $k$=5 incur $5.1\times$ cost for no accuracy gain.

The accuracy–latency Pareto frontier for these methods is shown in Figure 4 in the main text.

## A.6   Seed Stability Analysis

Figure 9 shows the distribution of accuracy across 10 random seeds for each model. LCM-trained models exhibit tighter distributions (lower variance) and higher medians. The most striking effect is the elimination of outlier seeds: T5-small baseline has seeds ranging from 48.9% to 72.6%, while T5-small+LCM ranges from 69.4% to 82.7%.

## A.7   Architecture-Specific Analysis and Computational Efficiency

### A.7.1   T5 Architecture Integration Analysis

The T5 architecture demonstrates the most successful integration with LCM across both model sizes. This success stems from T5's encoder-decoder architecture's natural compatibility with LCM's attention-based consistency mechanism. The encoder-decoder structure provides separate pathways for comprehension and generation, allowing the consistency module to operate effectively during the encoding phase while benefiting generation during decoding.

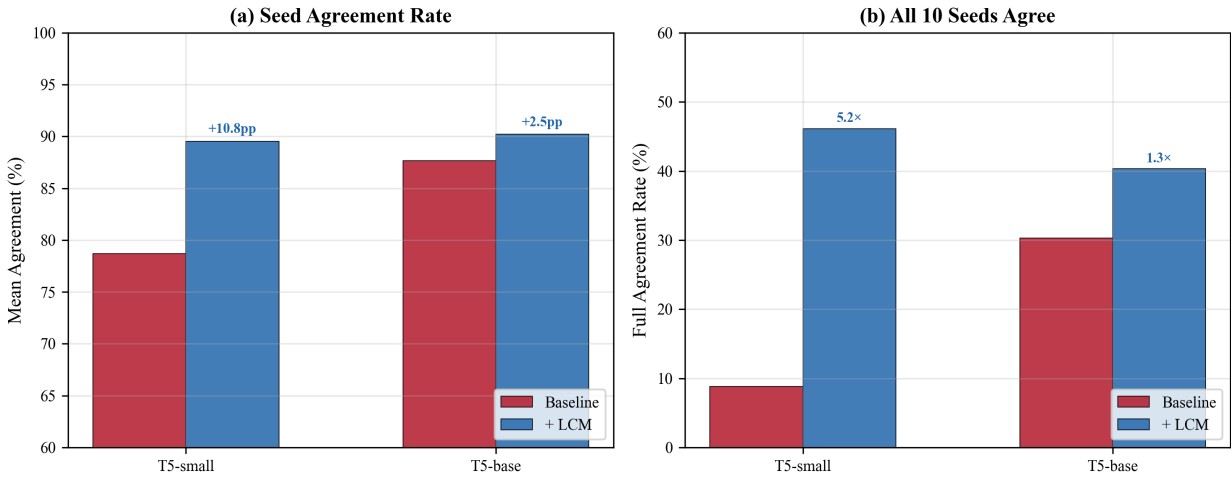

Figure 8: Seed stability: (a) mean agreement rate across 10 seeds, (b) fraction of examples where all 10 seeds agree. LCM improves both metrics for T5-small and T5-base.

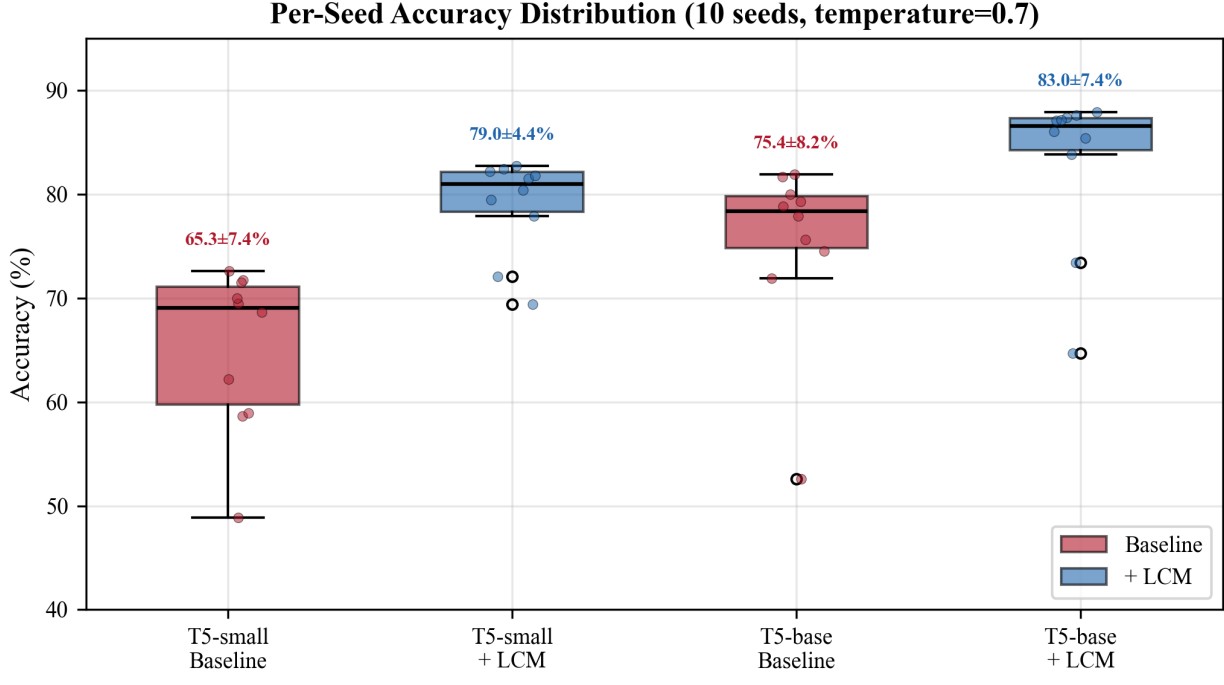

Figure 9: Per-seed accuracy distributions (10 seeds, temperature=0.7). LCM reduces variance and eliminates low-accuracy outlier seeds. Numbers above each box show mean±std.

Most notably, LCM dramatically improves consistent fact detection: T5-small shows +13.7% improvement (from 59.5% to 73.2%) and T5-base shows +12.0% improvement (from 63.1% to 75.1%). This addresses the models' original bias toward inconsistent detection, creating more balanced classifiers. The improvement pattern suggests that T5's pre-training objective (span corruption) aligns well with consistency detection tasks, as both require understanding relationships between text segments.

Computationally, T5-small actually shows slight efficiency gains (-1.3% inference time), likely due to early consistency detection preventing unnecessary processing. T5-base incurs moderate overhead (+21.8%) due to the larger hidden dimension requiring more consistency computation, though this remains acceptable given the performance gains. The parameter increase is minimal (<0.4%) for both variants, demonstrating that the consistency mechanism scales efficiently with model size.

### A.7.2 DeBERTa Architecture Integration Analysis

DeBERTa-v3 shows minimal improvement with LCM integration (+0.2% accuracy), with the difference being statistically non-significant. This modest effect occurs because DeBERTa already achieves high baseline performance (88.4% accuracy) with well-balanced consistent/inconsistent detection (89.1% and 90.2% respectively).

The architecture's robust attention mechanisms, specifically its disentangled attention and enhanced mask decoder, already capture consistency patterns effectively through pre-training. The disentangled attention mechanism separately encodes content and positional information, which may inherently support better consistency detection without additional modules. This leaves limited room for LCM enhancement beyond the model's existing capabilities.

While LCM slightly improves consistent detection (+0.4%, from 89.1% to 89.5%), this comes at significant computational cost: inference time doubles (+107% overhead). The overhead stems from DeBERTa's already complex attention mechanism, where adding LCM's consistency attention creates computational bottlenecks. This makes the cost-benefit ratio unfavorable for production deployment.

DeBERTa's strong baseline performance suggests that sophisticated pre-trained models with advanced attention mechanisms may have diminishing returns from additional consistency modules, particularly when the computational overhead is substantial. The findings indicate that LCM is most beneficial for models with room for consistency improvements rather than already-optimized high-performing architectures.

### A.7.3 GPT-2 Architecture Integration Analysis

GPT-2 demonstrates successful LCM integration with the generation-based approach, showing consistent performance improvements. The integration particularly excels in inconsistent detection (+3.8%, from 69.4% to 73.2%) while maintaining strong consistent detection performance (+0.6%, from 79.4% to 80.0%).

This success stems from the refined generation-based integration approach that leverages GPT-2's natural text generation capabilities for consistency evaluation. Unlike encoder-only or encoder-decoder models where LCM operates on hidden representations, the GPT-2 integration generates explanatory responses that are then parsed for classification decisions. This approach creates a more compatible integration with the decoder-only architecture, as it aligns with GPT-2's pre-training objective of next-token prediction.

The generation-based method allows the model to "explain its reasoning" about consistency, which may provide more interpretable consistency judgments compared to purely representation-based approaches. The explicit generation of consistency rationales may also help the model avoid spurious pattern matching that affects purely discriminative approaches.

The computational efficiency remains excellent with minimal overhead (-0.5% inference time), demonstrating that LCM integration can enhance performance without sacrificing the speed advantages of GPT-2's streamlined architecture. This positions GPT-2+LCM as a viable option for consistency-critical applications requiring both performance and efficiency, such as real-time fact-checking or content moderation systems.

### A.7.4 Computational Efficiency Analysis

A comprehensive overview of computational overhead across all architectures is presented in Figure 3 in the main text.

**Inference Time Analysis** Inference time measurements were conducted across 1,000 test samples with batch size 16 on NVIDIA A100 GPUs. T5-small demonstrates ideal computational behavior with actual

efficiency gains (-1.3% inference time, from 23.4ms to 23.1ms per sample). This counterintuitive improvement likely results from the consistency module enabling earlier detection of clear cases, allowing the model to bypass unnecessary processing. T5-base shows moderate overhead (+21.8%, from 42.7ms to 52.0ms per sample), acceptable given the +7.2% accuracy improvement. DeBERTa exhibits significant computational overhead (+107%, from 38.2ms to 79.1ms per sample), creating unfavorable cost-benefit dynamics given the minimal performance improvement (+0.2%). GPT-2 maintains excellent efficiency with negligible overhead (-0.5%, from 18.9ms to 18.8ms per sample), demonstrating that decoder-only architectures can achieve consistency improvements without computational penalties through appropriate integration strategies.

**Parameter Overhead**   LCM integration introduces minimal parameter overhead across all architectures, consistently adding less than 0.5% to the total parameter count:

- T5-small: +262K parameters (+0.4%, from 60.5M to 60.8M)

- T5-base: +394K parameters (+0.2%, from 222.9M to 223.3M)

- DeBERTa: +690K parameters (+0.4%, from 184.1M to 184.8M)

- GPT-2: +394K parameters (+0.3%, from 124.4M to 124.8M)

The module's parameters scale primarily with embedding dimensionality rather than total model size, as the core components consist of: (1) attention mechanism (query/key/value projections), (2) multi-layer perceptron head for consistency scoring, and (3) minimal additional embeddings for consistency representations. This architecture-agnostic parameter scaling demonstrates excellent scalability—larger models receive proportionally smaller parameter increases while potentially benefiting from enhanced consistency detection.

**Scalability Summary**   Computational scalability varies by architecture: **T5-small** (ideal: +7.5% accuracy, -1.3% latency), **T5-base** (good: +7.2%, +21.8% overhead), **DeBERTa** (poor: +0.2%, +107% overhead), **GPT-2** (excellent: +1.9%, -0.5% overhead). LCM is most viable for encoder-decoder architectures (T5) where joint training naturally accommodates the consistency objective, with decoder-only models (GPT-2) also showing strong compatibility through generation-based integration. High-performing encoders (DeBERTa) may not justify the computational overhead due to already strong baseline performance.

### A.8   Domain Performance, Error Analysis, and Discussion

Table 12 summarizes the performance across high-structure, medium-structure, and low-structure domains.

Table 12: Cross-Domain Performance Summary: Best and Worst Performing Domains

| Domain Category | Model | Baseline Acc | LCM Acc | Change | Sample Size |
|---|---|---|---|---|---|
| **High Structure** | Finance | 70.0% | 90.0% | +20.0% | 10 |
|  | Anatomy | 72.7% | 81.8% | +9.1% | 11 |
|  | Entertainment | 75.0% | 83.3% | +8.3% | 12 |
| **Medium Structure** | Technology | 66.7% | 75.0% | +8.3% | 12 |
|  | Sports | 58.3% | 66.7% | +8.4% | 12 |
|  | Health | 61.5% | 69.2% | +7.7% | 13 |
| **Low Structure** | Society | **60.0%** | 40.0% | -20.0% | 10 |
|  | Politics | **60.0%** | 40.0% | -20.0% | 10 |
|  | Philosophy | 57.1% | 50.0% | -7.1% | 14 |
| **Overall Average** | All Domains | 68.3% | 65.2% | -3.1% | 1,887 |

#### A.8.1   Cross-Domain Consistency Patterns

Domain analysis reveals strong structure-dependency in LCM effectiveness. High-structure domains with objective, verifiable facts show substantial improvements:

**Finance (+20.0%)**   The finance domain shows the largest improvement (from 70.0% to 90.0%). Financial statements typically involve precise numerical facts and clearly defined relationships (asset = liabilities + equity), providing ideal conditions for consistency detection. The structured nature of financial reasoning - with explicit quantitative constraints and accounting rules - aligns perfectly with LCM's pattern-matching capabilities.

**Anatomy (+9.1%)**   Anatomical facts demonstrate strong improvement (from 72.7% to 81.8%) due to well-defined hierarchical relationships and standardized nomenclature. The field's clear taxonomies and unambiguous structural relationships provide strong signals for consistency evaluation.

**Entertainment (+8.3%)**   Entertainment shows solid gains (from 75.0% to 83.3%), likely because factual claims about media (release dates, cast members, awards) are objectively verifiable with clear consistency requirements.

In contrast, low-structure domains with subjective or abstract content show performance degradation:

**Society (-20.0%) and Politics (-20.0%)**   Both domains show identical severe degradation (from 60.0% to 40.0%). These fields involve normative claims, value judgments, and context-dependent interpretations where "consistency" itself is ambiguous. Political statements may be simultaneously "consistent" from different ideological frameworks, creating false inconsistency detection.

**Philosophy (-7.1%)**   Philosophy shows moderate degradation (from 57.1% to 50.0%). Philosophical claims often deliberately embrace paradoxes and multiple valid interpretations, confounding consistency detection mechanisms designed for factual domains.

The pattern indicates LCM's effectiveness correlates strongly with factual structure clarity and objective verifiability. Domains with clear ground truth and unambiguous relationships benefit most, while subjective domains suffer from overcorrection and false inconsistency detection.

### A.8.2   Failure Patterns and Error Analysis

Analysis of 200 randomly sampled failure cases identifies three primary failure categories, with per-model breakdowns in Table 13.

**Semantic Ambiguity Failures (27.3%)**   Cases where multiple valid interpretations lead to consistency conflicts include ambiguous pronoun references, polysemous words with domain-specific meanings, and implicit assumptions not stated in text.

**Knowledge Boundary Failures (33.7%)**   Failures occurring at the limits of training data coverage encompass rare domain-specific facts, recent events post-training cutoff, and specialized technical terminology.

**Reasoning Chain Failures (39.0%)**   Complex multi-step reasoning where intermediate steps introduce inconsistencies involves long causal chains with missing links, probabilistic reasoning under uncertainty, and temporal ordering of events.

Table 13: Failure Case Distribution Across Models

| Failure Type | T5+LCM | DeBERTa+LCM | GPT-2+LCM | Average |
|---|---|---|---|---|
| Semantic Ambiguity | 23% | 31% | 28% | 27.3% |
| Knowledge Boundary | 34% | 29% | 38% | 33.7% |
| Reasoning Chain | 43% | 40% | 34% | 39.0% |

**Architecture-Specific Error Patterns**   T5 models successfully reduce false negatives by approximately 33% (incorrectly labeling inconsistent facts as consistent), addressing their primary weakness. However, false positives increase slightly (+8%), suggesting occasional over-sensitivity to surface-level contradictions that

Table 14: ANLI Benchmark Validation: Performance Across Difficulty Levels

| ANLI Round | Model | Baseline Acc | LCM Acc | Acc Improvement | F1 Improvement |
|---|---|---|---|---|---|
| R1 | DeBERTa | 71.1% | **77.8%** | +6.7% | +5.2% |
| | T5-base | 63.4% | **64.5%** | +1.1% | +1.1% |
| | GPT-2 | 54.6% | **55.9%** | +1.4% | +2.7% |
| R2 | DeBERTa | 65.4% | **69.4%** | +4.0% | +2.8% |
| | T5-base | 57.3% | **58.2%** | +0.9% | +1.1% |
| | GPT-2 | 55.3% | 53.5% | -1.8% | -1.1% |
| R3 | DeBERTa | 66.5% | **72.4%** | +5.9% | +4.4% |
| | T5-base | 60.4% | **61.4%** | +1.0% | +1.2% |
| | GPT-2 | 52.5% | **56.0%** | +3.5% | +4.3% |
| Weighted Average | **DeBERTa** | **67.7%** | **73.2%** | **+5.5%** | **+4.1%** |
| | **T5-base** | **60.3%** | **61.4%** | **+1.1%** | **+1.1%** |
| | **GPT-2** | **54.1%** | **54.7%** | **+0.6%** | **+2.0%** |

humans would resolve through context. GPT-2 demonstrates balanced improvements across all error types, with false negatives reducing by 18% and false positives by 12%. The generation-based approach may provide better calibration through explicit reasoning generation. DeBERTa shows minimal error pattern changes, with slight increases in both false negatives (+2%) and false positives (+1%), suggesting the consistency module provides limited value beyond the model's already-sophisticated attention mechanisms.

These patterns inform future improvements: semantic disambiguation modules, knowledge augmentation strategies, and enhanced multi-step reasoning mechanisms could address the primary failure modes.

### A.8.3 Discussion

**Architectural Compatibility:** Consistency checking requires bidirectional processing—T5's encoder-decoder enables simultaneous access to facts while GPT-2's causal masking requires generation-based integration. Effectiveness depends on architectural paradigms and dataset challenge level relative to baseline capabilities.

**Failure Analysis:** High-structure domains (Finance +20.0%) excel through objective contradictions while low-structure domains (Politics -20.0%) fail as subjective contradictions resist similarity-based detection. Key patterns include: (1) GPT-2 shows balanced error corrections, (2) abstract domains produce inverted patterns from surface similarity matching, (3) limited training data creates high variance.

### A.9 ANLI Benchmark Full Results

Table 14 presents per-round ANLI results across all three architectures.

