# OpenReview forum: "Lightweight Consistency Memory: Training-Time Regularization for Logical Consistency in Small Language Models"
_TMLR — Under review for TMLR_

### Review · Reviewer_J4cJ · 2026-06-11

**Summary Of Contributions:**

The paper introduces Lightweight Consistency Memory (LCM), an auxiliary module for consistency checking that is trained jointly with a backbone language model and then discarded at inference for encoder–decoder and decoder-only backbones (T5 and GPT-2), leaving the backbone’s inference-time latency unchanged. The LCM head embeds each background fact and the candidate into a shared embedding space, uses cosine-similarity soft attention to compute a relevance-weighted aggregation over background facts, and feeds the concatenation of candidate embedding and aggregated background embedding into an MLP classifier. The computational complexity is $O(nd)$ in the number of facts $n$ and embedding dimension $d$. The authors apply LCM to T5-small, T5-base, GPT-2, and DeBERTa-v3, and evaluate on consistency-targeted benchmarks comprising 18,862 examples over 156 topics plus ANLI-style adversarial NLI data.

**Strengths**

1. The focus on training-time consistency signals that are amortized into the backbone weights, with zero (T5, GPT‑2) or controlled (DeBERTa) inference cost, is well argued and highly relevant to practitioners who care about both reasoning reliability and latency budgets.
2. The design is plausibly easy to integrate into existing training pipelines and backbones, and the paper demonstrates cross-paradigm applicability (encoder–decoder, decoder-only, encoder-only).
3. LCM improves mean agreement across seeds and reduces variance, eliminating low-accuracy outlier seeds for T5-small and T5-base. The results also suggest that the auxiliary consistency supervision stabilizes training dynamics, which is interesting from an optimization perspective.

**Weaknesses**

1. LCM is a supervised auxiliary head that learns to perform NLI-style consistency classification and injects its gradients into the backbone during fine-tuning. Compared to methods like paraphrase-based consistency fine-tuning, consistency reward models, logic-loss penalties against external KBs, the main novelty seems to be moderate.
2. It remains unclear how much of the observed gain translates into improvements on *downstream* tasks where consistency is beneficial but not directly supervised, such as multi-hop QA, fact verification in longer documents, or instruction following with latent contradictions.
3. I do not see a detailed ablation that disentangles which components drive the gains. This includes cosine vs dot-product attention, a simple pooled-background baseline, using only candidate embeddings or removing the auxiliary loss schedule $\alpha(t)$.
4. The DeBERTa-v3 experiments suggest that for already-strong models with sophisticated attention mechanisms, LCM yields minimal gains on standard benchmarks while significantly increasing inference cost when kept active, due to added attention computations.

**Audience:**

Yes

**Audience Explanation:**

1. This is a well-motivated and clearly written paper that proposes a lightweight training-time consistency module for small language models, with solid empirical gains on targeted consistency benchmarks and useful analysis.
2. The results show that a relatively small auxiliary signal can, in some cases, bring a 60M-parameter model close to a much larger DeBERTa-v3 NLI classifier (184M parameters, ~80.6% accuracy), which is an attractive result for edge or embedded deployments.

**Broader Impact Concerns:**

No.

**Claims And Evidence:**

Yes

**Claims Explanation:**

1. $O(nd)$ complexity and <0.5% parameter overhead when used with T5/GPT‑2.
2. The gains on T5-small and T5-base are sizable (+7.5% and +7.2% accuracy on consistency benchmarks), and the authors emphasize that these gains are achieved at effectively unchanged inference latency because the LCM module is discarded after training.

**Requested Changes:**

1. Given the already rich literature on training-time consistency and memory-augmented modules, is there a theoretical or qualitative reason why this particular geometric design should generalize better than other auxiliary heads?
2. It is not fully clear how the main consistency dataset is constructed. Are facts sampled from existing NLI corpora, from Wikipedia, or synthetic templates? How are background sets assembled, and how are contradictions generated (negation, causal inconsistencies, numeric mismatches, etc.)?
3. How does LCM compare to a simpler baseline where the backbone is simply multi-tasked on LM and NLI using a shared encoder and a standard linear classifier head, but without the explicit cosine-attention memory component?
4. Is there any evidence that LCM harms language modeling or general downstream performance when the auxiliary loss is too strong, and how did you select the final settings reported?

---

### Review · Reviewer_xo2C · 2026-07-13

**Summary Of Contributions:**

The paper proposes Lightweight Consistency Memory (LCM), an auxiliary consistency objective jointly trained with T5, GPT-2, and DeBERTa. The auxiliary head is removed at inference for T5 and GPT-2, while DeBERTa retains it through a fusion layer. The evaluation covers a synthetic consistency dataset, ANLI, architectural ablations, sampling stability, and inference cost.

**Strengths**

1. The paper addresses a practical problem: improving consistency in small models without relying on expensive multi-pass inference.
2. The evaluation spans encoder-decoder, decoder-only, and encoder-only architectures, and includes an external benchmark rather than relying entirely on the authors’ synthetic data.
3. The paper provides useful implementation details, ablations, generation prompts, latency measurements, and some discussion of architecture- and domain-dependent failure cases.

**Weaknesses**

1. The dataset and evaluation protocol are not sufficiently validated. The dataset is generated and labeled by LLMs without reported human verification, and the train/test split does not appear to separate source paragraphs or topics. More importantly, the task is defined as binary although the dataset contains neutral examples, and the reported evaluation sizes vary between 996, 1,688, and 1,887 without a clear explanation.
2. The experiments do not isolate the contribution of the proposed LCM architecture from ordinary auxiliary supervision. A backbone trained with the same consistency labels, updates, and compute using a simple auxiliary classifier is missing. Mean pooling is also not significantly worse than full LCM, suggesting that the proposed attention mechanism may contribute little to the reported gains.
3. Several statistical and efficiency claims are internally inconsistent. The results are described as single-run estimates, but the construction of the paired tests and effect sizes is unclear. GPT-2 is reported as significant at p<0.05 even though the stated Bonferroni threshold is 0.004. The zero-overhead claim also conflicts with the reported +21.8% T5-base latency, while the explanation based on “early consistency detection” cannot apply when the LCM branch has been removed.

**Additional Comments:**

1. How are neutral examples handled in each experiment, and what causes the test sizes of 996, 1,688, and 1,887? Which exact subset supports each headline result?
2. Were the standalone and LCM models trained with identical data, numbers of backbone updates, early-stopping criteria, and hyperparameter-search budgets? How does a supervision-matched auxiliary MLP baseline perform?
3. How can a discarded LCM branch produce +21.8% T5-base latency or enable early termination? Were output lengths, decoding settings, warm-up runs, and model graphs identical during timing?

**Audience:**

Yes

**Audience Explanation:**

Training-time consistency supervision with no additional inference module is relevant to researchers working on efficient language models, multi-task learning, and model reliability. A carefully controlled version of this study could provide useful evidence about when auxiliary consistency objectives transfer into backbone representations.

**Broader Impact Concerns:**

I do not see a distinct broader-impact issue requiring a separate statement. However, claims about use in medical devices, fact-checking, or content moderation should be moderated until reliability has been validated on human-annotated and naturally occurring data.

**Claims And Evidence:**

No

**Claims Explanation:**

The main claim is that the specific LCM design produces reliable consistency improvements while preserving native inference cost. At present, dataset uncertainty prevents a reliable assessment of generalization, the missing supervision-matched controls prevent attribution of the gains to LCM, and the statistical and latency inconsistencies weaken the quantitative conclusions. The idea remains plausible, but the current evidence does not distinguish it from standard consistency-task fine-tuning.

**Requested Changes:**

1. Clarify and audit the dataset pipeline. Report human label validation, split data by source paragraph or topic, specify exactly how neutral samples are handled, and use one consistent evaluation population across all main tables.
2. Add equal-data, equal-update, and equal-hyperparameter controls, including direct consistency fine-tuning and a plain auxiliary classifier without the proposed attention-based aggregation. Report results over multiple training seeds.
3. Recompute the statistical analysis and inference measurements. Use tests appropriate for paired predictions, apply the stated multiple-comparison correction consistently, and reconcile the zero-overhead claim with the reported T5 latency differences. The complexity discussion should also include the cost of obtaining separate fact and candidate representations, not only the auxiliary head.

---

### Review · Reviewer_CWnf · 2026-07-13

**Summary Of Contributions:**

This paper proposes an auxiliary loss for fine-tuning language models on NLI.


**Strengths:**
- I like the ANLI out-of-distribution evaluation. It is a good practice, but it could be highlighted more.
- The ablation study is a nice addition and seems to be well-designed: four variants each removing exactly one component, McNemar tests, plus training-dynamics analysis explaining why the MLP is necessary.
- Cross-architecture coverage, evaluating this method on encoder-decoder (T5), decoder-only (GPT-2) and encoder-only (DeBERTa) language models.

**Weaknesses:**
- The scope is much narrower than advertised. Initially, the paper talks about general language modeling and logical consistency, but it actually only applies the method to fine-tuning existing models on a single small LLM-generated NLI dataset.
- Even if we frame the paper as a new auxiliary loss for improving NLI finetuning, evaluating the method only on a single small LLM-generated (without any manual validation) dataset is not very convincing. I appreciate also including an OOD test on ANLI! But the results would look much stronger if the authors trained models on well-established datasets and compared to reference results from previous work.
- The paper is not written very clearly, it repeats several claims multiple times and overwhelms the reader with unnecessary details while failing to properly describe the important details at the same time. It is not clear what exactly the D and D_cons datasets are, it is not clear why DeBERTa uses a different training methods, how are GPT answers parsed and how are they generated for supervised training? Why is *n* different in Table 2 (1887), Table 4 (1688) and Table 7 (996 for T5-small)? How are the neutral labels used for training BCE? Is ANLI tested as a zero-shot transfer of the fine-tuned model?
- I find the name "Lightweight Consistency Memory" very confusing as there is no memory (the authors themselves claim that it is "stateless"), the method only adjusts semantic embeddings gathered from a language model via cosine similarity.
- The proposed method uses the best training setting after an extensive hyperparameter search (Appendix A.3.5), but it seems that 1) this hyperparameter search has not been applied to the baselines, 2) the baselines use the hyperparameters optimized for the proposed method. That is methodologically wrong. Furthermore, details about the baseline training are missing in general, it's not even clear if it has been trained on comparable amounts of data or compute -- LCM models get a 25+ epoch three-phase schedule, are the baseline models trained comparably?
- Contradiction? "T5 and GPT-2 retain native latency because LCM is discarded at inference" yet Figure 3/A.7.4 report +21.8% inference overhead for T5-base, explained as "the larger hidden dimension requiring more consistency computation" — which is impossible if LCM is absent at inference. Similarly, T5-small's −1.3% latency is attributed to "early consistency detection preventing unnecessary processing" (A.7.1) — again impossible for a discarded module.
- If I understand the description correctly, all main results are single training runs and their own analysis shows extreme sensitivity (baseline T5-small ranges 48.9–72.6% across different decoding seeds). Significance is computed over test examples with the trained model treated as fixed; training-seed variance is never measured.


**Minor comment:**

The leading example in the introduction is this:

> For example, given the background fact “beta-blockers reduce heart rate,” a small model may judge the statement “beta-blockers are recommended for tachycardia” as consistent—failing to detect the logical contradiction.

But as far as I know, tachycardia means abnormally rapid heart rate, so I don't see any logical contradiction.

**Audience:**

Yes

**Audience Explanation:**

The topic -- improving logical consistency of small (60–220M) models via a training-time auxiliary objective rather than multi-pass inference -- is relevant to readers working on efficient NLP and resource-constrained deployment. Several findings would interest this audience *if credibly established*: that self-consistency and best-of-N fail to help (or hurt) small models, that auxiliary consistency training substantially improves output stability across seeds, and that gains for strong models appear only when dataset difficulty exceeds their capability (DeBERTa: +0.2% in-house vs. +5.5% ANLI). My concerns are about the support for the claims (see above), not the relevance of the questions.

**Broader Impact Concerns:**

No concerns.

**Claims And Evidence:**

No

**Claims Explanation:**

Several main claims are contradicted by the paper's own numbers, others rest on flawed methodology (see weaknesses):

1. "Zero inference overhead" -- contradicted by Figure 3/A.7.4, which report +21.8% overhead for T5-base and −1.3% for T5-small with mechanistic explanations that are impossible if LCM is discarded at inference.
2. Only single training runs are p-valued despite the paper's own evidence of extreme seed sensitivity (48.9-72.6% just across different decoding); baselines apparently not given comparable hyperparameter tuning.
3. "Matches DeBERTa-v3 with 60M params* -- the paper reports two irreconcilable numbers for DeBERTa fine-tuned on the same data (88.4%, Table 1 vs. 80.6%, §5.3) and claims parity against the weaker one.
4. Inconsistent evaluation -- n and scores shift across tables without explanation (T5-small+LCM: 79.2/79.5/82.2 in Tables 1/2/3), and the abstract mixes numbers from different protocols.
5. Unvalidated evidence base -- train/validation/test data are LLM-generated with no human validation; the paper's own motivating example is mislabeled (beta-blockers *treat* tachycardia).

**Requested Changes:**

*Critical to securing my recommendation for acceptance:*

1. **Reconcile the internally contradictory numbers.** Explain the +21.8%/−1.3% latency changes for T5-base/T5-small (Fig. 3, A.7.4) given that LCM is discarded at inference; the two DeBERTa results on the same dataset (88.4% Table 1 vs. 80.6% §5.3); Table 12's overall −3.1% vs. Table 1's +7.5%; and the shifting n and scores across Tables 1/2/3/7 (e.g., T5-small+LCM: 79.2/79.5/82.2).
2. **Report multi-seed training runs** (mean±std over ≥3–5 training seeds) for the main comparisons, with significance computed over runs rather than treating a single trained model as fixed.
3. **Fair baselines.** Apply comparable hyperparameter tuning and training compute to the standalone baselines, and document the baseline training protocol.
4. **Validate the dataset.** Human-audit a random sample of labels and report agreement, and add a hypothesis-only (candidate-only) baseline to rule out annotation artifacts. Fix or replace the mislabeled beta-blocker example in the introduction.
5. **Scope the claims to what is shown:** an auxiliary classification loss for fine-tuning on a synthetic NLI-style task, not general logical consistency of language models.

*Would strengthen the work:*

6. Specify the missing experimental details: composition of D vs. D_cons, GPT-2 answer generation/parsing, handling of neutral labels in the BCE loss, and the ANLI protocol (zero-shot transfer? how does a binary model produce 3-way predictions?).
7. Compare against at least one cited training-time consistency method (e.g., Chain of Guidance) or a simple alternative such as MNLI multi-task fine-tuning, rather than only inference-time methods (different class).
8. Rename the method.
9. Guard against train/test leakage by splitting on source paragraphs rather than stratifying by topic/label.
10. Tighten the writing: remove the repeated O(nd)/"discarded at inference" claims and move key protocol details from the appendix into the main text.